

# Heterologous expression of the *Stellaria media* plant defensin SmD1 in *Escherichia coli*

Yiyi Qiu[1] and Qiaozhi Song[2,3]

[1] Zhejiang Institute of Economics and Trade, Hangzhou, China
[2] Institute of Food Science and Technology CAAS/Comprehensive Utilization Laboratory of Cereal and Oil Processing, Ministry of Agriculture and Rural, Beijing, China
[3] School of Chemical Engineering and Technology, Tianjin University, Tianjin, China

## ABSTRACT

SmD1 isolated from seeds of common chickweed *Stellaria media* has strong inhibitory activity against phytopathogenic fungi and oomycetes in the micromolar range ($IC_{50} \leq 1$ μM). However, the low production of plant defensins in natural strains limits their large-scale actual production. In this study, defensin gene *SmD1* was successfully heterologously expressed in *Escherichia coli* BL21 (DE3) for efficient production of plant defensins. The defensin gene *SmD1* fused with thioredoxin was cloned into pET22b (+) vector. Then, it was transformed into *E. coli* BL21 (DE3) and expressed solubly after induction of isopropyl-β-D-thiogalactopyranoside (IPTG). At 50 °C, active SmD1 was released by 50% (v/v) formic acid hydrolysis of the cleavage of Asp-Pro bond between fused proteins. The recombinant protein SmD1 was purified by Ni-IDA column and showed significant antifungal activities against fungi. The induction conditions was optimized, and the results showed that the antimicrobial activity reached its maximum when the IPTG had a concentration of 0.6 mmol/L, a temperature of 25 °C, an induction time of 12 h and an $OD_{600}$ of 0.8.

## INTRODUCTION

Plant defensins, as a cysteine-stabilized antimicrobial peptide, is a fundamental component of innate immunity in plants and serves as the first line of defense against pathogenic microbial invasion (*Zhao et al., 2011*). The defense component of plants includes phytoalexins and phytoanticipins, PR-proteins, enzyme inhibitors, and antimicrobial peptides (AMPs) (*Darvill & Albersheim, 1984*; *Selitrennikoff, 2001*; *Sels et al., 2008*; *Lima et al., 2022*). Plant defensin is a small cationic AMP rich in cysteine, which is a basic component of the plant defense system. Defensins have activity against a range of pathogenic organisms at low concentrations and can be used as novel germicides to prevent plant diseases (*Koike et al., 2002*; *Van Loon, Rep & Pieterse, 2006*; *Li et al., 2021*; *Ali et al., 2018*; *Leannec-Rialland et al., 2022*; *Slezina & Odintsova, 2023*). Compared with antibiotics, their antimicrobial activities have great diversity due to being unaffected by common mutations that lead to antibiotic resistance (*Browne et al., 2020*; *Nazarian-Firouzabadi, Torres & Fuente-Nunez, 2024*). Most of the plant defensin isolated is

Corresponding author
Qiaozhi Song, songgao241@163.com

extracted from seeds. Defensins interact with fungal-specific lipid components in the plasma membrane. For example, the high-affinity binding sites for plant defensins DmAMP1 and RsAFP2, isolated from *Dahlia merckii* and *Raphanus sativus*, respectively (*Ramamoorthy et al., 2010*; *Thevissen et al., 2003*) induced an array of relatively rapid responses in fungal cells, including increased potassium efflux and calcium uptake, as well as membrane-permeabilization (*Thevissen et al., 1996*; *Thevissen, Terras & Broekaert, 1999*; *Oliveira et al., 2022*). However, the industrial application of plant defensins has long been constrained by keybottlenecks of low abundance in natural sources.

As short polypeptides, defensins could be obtained throughisolation from natural sources and chemical synthesis. Due to low yield and complex impurities, chemical synthesis requires several setps to prepare peptide fragments through their isolation/purification, which is often uneconomical for long residue peptides (*Gong et al., 2024*). In particular, plant defensin has four disulfide bonds as post-translational modifications that are not conducive to its molecular synthesis (*Shanmugaraj et al., 2021*). Now numerous biological expression systems have been introduced for the economical production of antimicrobial peptides (*Kordi et al., 2024*). Recombinant peptide production is a relatively cheap and simple to manipulate alternative method (*Nazarian-Firouzabadi, Torres & Fuente-Nunez, 2024*; *Li et al., 2023*).

Nevertheless, the recombinant route is limited by peptide toxicity to host cells, proteolysis of products, and sometimes poor yields (*Lee et al., 2000*; *Wang et al., 2008*; *Deo et al., 2022*). *E. coli* is one of the major systems used for producing recombinant antimicrobial peptides and accounts for more than 80% of all cases (*Li & Chen, 2010*; *Lappöhn et al., 2023*). As a heterologous expression system, *E. coli* has a high density of cells and can yield high intracellular expression (*Rosano, Morales & Ceccarelli, 2019*). Various fusion carriers such as thioredoxin (TRX) (*Satei et al., 2021*), glutathione transferase (GST) (*Di Somma et al., 2021*; *Shendge & D'Souza, 2022*), N[pro] (an autoprotease from the classical swine fever virus) (*Achmüller et al., 2007*), amidophosphoribosyltransferase (PurF) (*Lee et al., 2000*) and small ubiquitin-related modifier (SUMO) (*Xu et al., 2023*) have been used for AMP expression and purification.

SmD1 isolated from seeds of common chickweed *Stellaria media* displays strong inhibitory activity against phytopathogenic fungi and oomycetes in the micromolar range ($IC_{50} \leq 1$ μM). The *SmD1* gene show promise for engineering pathogen resistance and expand our knowledge on weed genomics (*Slavokhotova et al., 2010*). In this study, we cloned the SmD1 defensin-encoding gene from *Stellaria media*. The plant defensin SmD1 and TRX were expressed in *E. coli* using a TRX-SmD1 fusion system. The defensin was produced in a prokaryotic system, purified, and evaluated for its antifungal activity *in vitro*. This system provides a universal solution for the large-scale production of plant defensins and also serves as a template for engineering other cysteine-stabilized antimicrobial agents.

## MATERIALS AND METHODS

### Microbial strains, culture media and enzymes

*E. coli* DH5α and *E. coli* BL21 (DE3) were grown in Luriae-Bertani (LB) medium at 37 °C. The indicator strains used in the plant defensin SmD1 were *Botrytis cinereal*

CGMCC3.4584, *Fusarium graminearum* CGMCC3.6862, *Fusarium oxysporum* CGMCC3.2830, *Rhizoctonia solani* CGMCC3.288. All these indicator strains were grown in PDA medium at 30 °C. The plasmid pET22b (+) was used as the expression vector and the plasmid pET32a (+) was used as template for amplification of TRX. DNA restriction enzymes, Taq DNA polymerase, PFU polymerase, T4 DNA ligase, were purchased from Transgen Biotech (Beijing, China). Other chemicals are of analytical grade and commercially available.

## Construction of expression plasmid

According to relevant literatures (*Slavokhotova et al., 2010*), a 153 bp mature plant defensin mature peptide sequence was synthesized by Shanghai Shenggong Bioengineering Technology. The DNA and protein sequences of TRX-SmD1 are shown in Fig. 1. The genes coding for SmD1 and the fusion partner TRX were amplified by PCR with two pairs of synthetic oligodeoxyribonucleotides TF (5′-CGCCCATGGCCATGAGCGATAAAATT ATTC-3′) and TR (5′-ATGATGATGATGATGATGGGCCAGGTTAGCGTCGAGG-3′), SF (5′-CATCATCATCATCATCATGATCCGAAGATCTGCGAACGTGC-3′) and SR (5′-CGCCTCGAGTTAGCAGTTGAAGTAGCAG-3′), respectively. *NcoI* and *XhoI* restriction sites were introduced into the primers TF and SR, respectively. The PCR products were purified using the Gel Extraction Kit (Tiangen, Beijing, China). The *SmD1* and *TRX* genes were mixed at the ratio of approximately 1:1, and then randomly ligated using *Pfu* polymerase. A PCR procedure was conducted with primers SR and TF and fused genes as the template. The PCR products were separated by agarose gel electrophoresis, and the expected DNA band was excised from the gel and purified with a DNA gel extraction kit (Tiangen Biotech, Beijing, China).

The fused gene was digested and purified uisng *NcoI* and *XhoI*, and then ligated into linearized plasmid pET22b (+) using T4 DNA ligase and the same restriction enzymes. The recombinant plasmid pET22b (+)-TRX-SmD1 was sequenced uisng GENEWIZ Biological Technology and then transformed into competent *E. coli* BL21 (DE3) cells.

## Induction expression of the fusion protein and expression position

SmD1 was expressed as a fusion protein in *E. coli* BL21 (DE3). For protein expression, the overnight culture of *E. coli* BL21 (DE3) harboring recombinant plasmid was inoculated into LB broth containing ampicillin (50 μg/mL) and incubated at 37 °C with agitation. The culture was induced with isopropyl-β-D-thiogalactopyranoside (IPTG) when the $OD_{600}$ reached 0.6, and further cultivated. The culture supernatant and the *E. coli* cell pellet were collected by centrifugation at 4,000 *g* for 20 min, and subjected to sodium dodecyl sulfate-polyacrylamide gel electrophoresis (SDS-PAGE). *E. coli* cells collected from the culture were resuspended in PBS and pelleted again. The cells were resuspended in cold 20% sucrose/30 mM Tris-HCl buffer (pH 8.0). This sample was saved as the whole-cell fraction. 1 M EDTA (pH 8.0) was added into the remainder of the sample. The sample was kept on ice for 10 min and then centrifuged at 4,000 *g* for 8 min. Then the supernatant was collected as the periplasmic fraction, and the pellet was the intracellular fraction. The cells

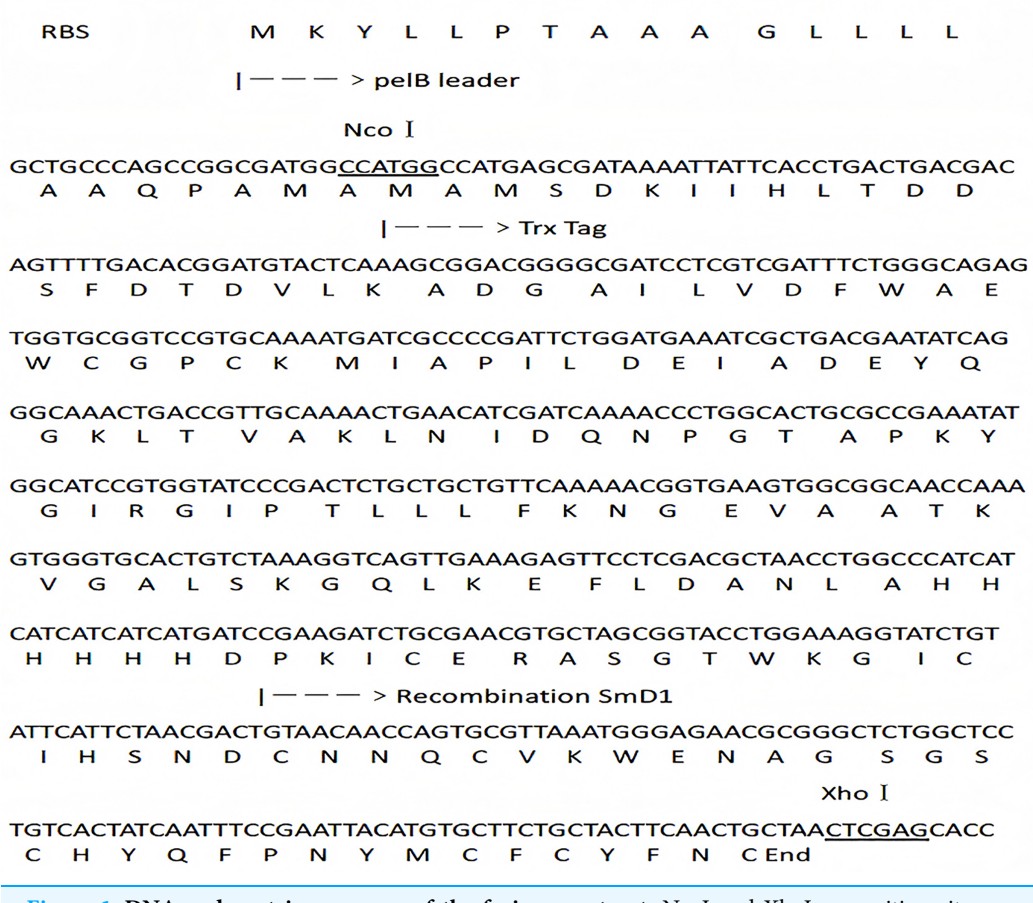

**Figure 1 DNA and protein sequence of the fusion construct.** NcoI and XhoI recognition sites are underlined. The sequence of pelB leader, TRX tag and SmD1 are labeled.

were collected and disrupted by sonication (SCIENJZ-IID, Ningbo, China) using 270 HZ amplitude. All these fractions were sampled for SDS-PAGE assay.

## Purification and cleavage of the fusion peptide

The signal peptide pelB and the fusion partner TRX with 6× His Tag are the binding part of Ni-IDA column used for secretion and purification. The soluble fraction after centrifugation was applied by $Ni^{2+}$-chelating chromatography. After extensive washing the contaminating proteins with binding buffer containing 10 mM imidazole, the target recombinant peptide was eluted with an elution buffer containing 160 mM imidazole at a flow rate of 1 mL/min. The peak fractions containing the fusion peptide were dialyzed successively against distilled water and concentrated with a 3 kD Centrifugal Filter Unit (Millipore, Burlington, MA, USA). After treated with 50% (v/v) formic acid in 50 °C for 72 h, the reaction mixture was then lyophilized for formic acid removal and analyzed by SDS-PAGE after dissolved in $ddH_2O$. Protein concentration was measured by the Bradford method with bovine serum albumin (BSA, Southampton, NY, USA) as a standard.

## Optimization of induction condition

The diameter of the inhibition zone against *F. graminearu* CGMCC 3.6862 was measured as a test indicator. Optimization of induction condition is based on the diameter of the inhibition zone. Induction conditions were optimized by varying the optical density ($OD_{600}$ = 0.4, 0.6, 0.8, 1.0 and 1.2), time (4, 6, 8, 10 and 12 h), temperature (16, 25, 30 and 37 °C), and IPTG concentration (0.2, 0.4, 0.6, 0.8, 1.0 and 1.2 mM) to obtain the strongest antifungal activity.

## Antifungal activity assay

The final purified plant defensin was stored in −20 °C and subsequently used to investigate the antifungal activity. The antifungal activity of the liberated SmD1 was tested by agar well diffusion assay using the indicator strains. The 50 μL culture was inoculated into 5 mL fresh PDA medium, and then the 80 μL of cell suspension was inoculated into 10 mL of preheating PDA medium containing 1.0% (w/v) agar. The medium was rapidly mixed and poured into the petri dish to form a uniform layer. The oxford cups (8 mm diameter) were put on the surface of the gelated medium. A total of 100 μL purified protein sample was added to oxford cup and then incubated at 30 °C for 48 h for measuring the diameter of the inhibition zone.

The minimum inhibitory concentration (MIC value) was determined by the double dilution method (*Li et al., 2016*). In a 96-well flat-bottomed microtiter plate, 90 μL of fungal suspensions (approximately $2 \times 10^6$ CFU/mL) was added. Meanwhile, 10 μL SmD1 with different final concentrations (0, 0.125, 0.25, 0.5, 1, 2, 4, 8, and 16 μg/mL) were also added, respectively. After incubation at 30 °C for 16 h, the MIC was determined by measuring the absorbance at 600 nm using a Synergy HTX Multi-Mode Reader. PDA medium was used as negative control and PDA medium containing fungi was used as positive control.

# RESULTS AND ANALYSIS

## Construction of expression plasmid

The expected size of the PCR amplified products of the gene encoding *SmD1* and the gene encoding *TRX* were 186 and 359 bp, respectively. The expected size of the *TRX-SmD1* gene overlap PCR product was 524 bp. The genes coding for the thioredoxin-*SmD1* gene were amplified (Fig. 2A). The colony PCR product was identified by 1.5% agarose electrophoresis to verify the insertion of the target fragment into the pET22b (+) vector (Fig. 2B). Sequencing of the recombinant plasmid confirmed that it was correctly constructed.

AMPs are often expressed by being fusing with a partner in heterologous hosts to neutralize their innate toxic activity and increase their expression levels (*Hoelscher et al., 2022*). In particular, thioredoxin gene fusion system is one of the successful systems in producing correctly folded and soluble heterologous protein in *E. coli* (*Saffari et al., 2020*). In this study, TRX was selected as a fusion partner because SmD1 is a small peptide that is susceptible to hydrolysis by proteases. TRX may also promote the formation of the correct structure of fusion proteins and reduce the toxic effects of recombinant peptides on the

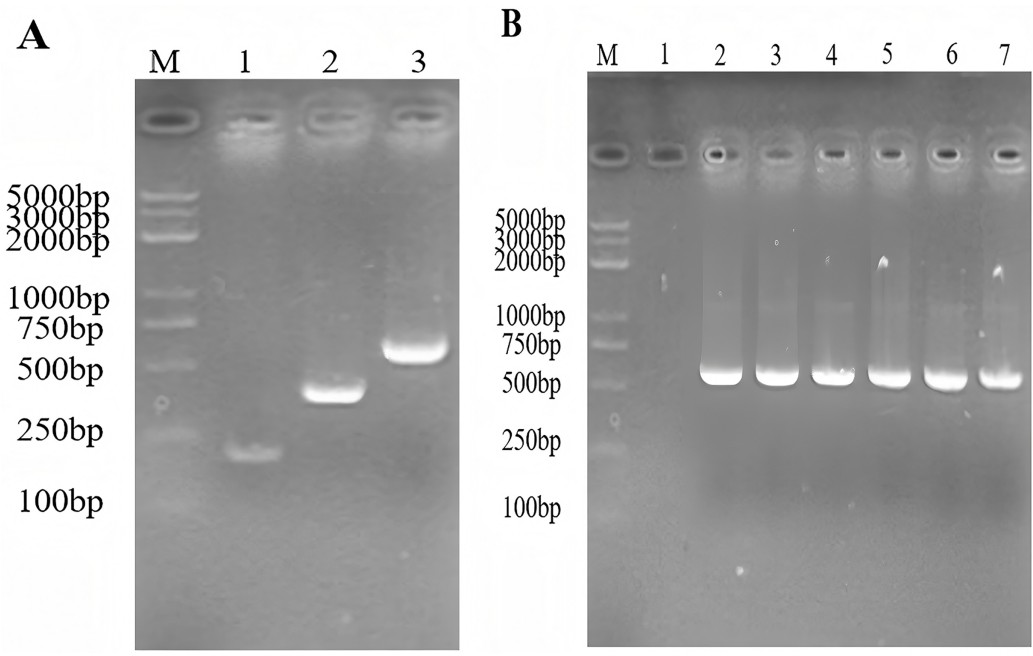

**Figure 2** (A) The construction of the TRX and SmD1 gene fusion using the overlap extension- PCR. M: standard DNA marker; 1–3: the amplified fragment of the TRX, the SmD1, and the TRX-SmD1 fusion gene. (B) Identification of the colony PCR product. M: standard DNA marker; colony PCR products of TRX-SmD1.

host cell (*Liu et al., 2013*) and the formation of inclusion bodies. The recombinant protein was purified by Ni-IDA column, and the fusion portion with a 6× His tag will bind to the Ni column. A D-P bond cleavage site was added between the plant defensin gene *SmD1* and *TRX* gene. The plasmid pET22b (+) was used as fusion expression system, which is a potential periplasmic expression vector with a pelB signal sequence. The fused protein Trx-SmD1 was cloned into the pET22b (+) and overexpressed under the induction of IPTG.

## Induction expression of the fusion protein and expression position

With IPTG induction, the fusion proteins TRX-SmD1 isolated from different fractions were subjected to SDS-PAGE analysis to locate the expression position (Fig. 3). The clear band at 21 kDa was detected as shown in band 4, corresponding to the expected size of the target protein. The clear band at 21 kDa was also detected in lane 3. The comparison of the periplasmic and intracellular protein pattern shows that *E. coli* kept a detectable amount of TRX-SmD1 in the periplasm and more fusion proteins were obtained in intracellular protein patterns. Therefore, with the help of the pelB signal peptide, a small part of the fusion proteins were transported across the inner membrane and secreted into the periplasmic space. It can be seen that the recombinant protein mainly exists in the form of a soluble fusion protein. There are no lanes in lane 2, indicating that the fusion proteins are not secreted outside the cell.

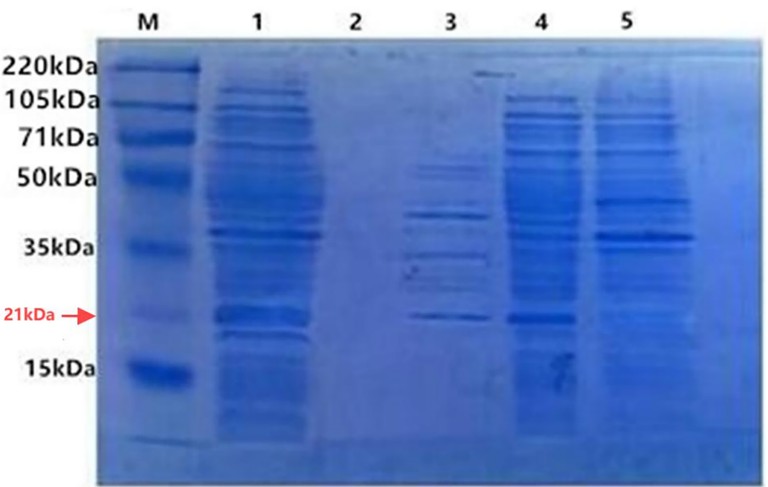

**Figure 3 SDS-PAGE of the fusion protein in recombinant *E. coli* cells isolated from different fractions with IPTG induction.** M, polypeptide molecular mass standards; Lane 1, the whole-cell proteins; Lane 2, extracellular supernatant proteins of culture; Lane 3, periplasmic proteins; Lane 4, intracellular soluble proteins; Lane 5, intracellular insoluble proteins.

## Purification and cleavage of the fusion peptide

*SmD1* was expressed in *E. coli* after IPTG induction as a full-length recombinant protein (21 kDa in size) with the fusion protein. The sonicated supernatant showed a significant band of the expected size on a gel (Fig. 4). This suggested that the recombinant SmD1 protein could be soluble within the cell. Purification by Ni-NTA affinity chromatography showed that most of the contaminating proteins had been washed away by the binding buffer with 10 mM imidazole, and the recombinant protein was concentrated.

After treated with 50% (v/v) formic acid in 50 °C for 72 h, the recombinant protein was cleaved into two parts. SDS-PAGE was used to determine the efficiency of purification and cleavage (Fig. 4). The size of TRX and the genetic engineering product SmD1 were about 15.2 and 5.8 kDa, respectively, indicating a successful specific cleavage by the hydrolyzation of D-P peptide bond through addition of formic acid.

SmD1 was easily released from its fusion partner after treated with 50% (v/v) formic acid and showed significant activities against phytopathogenic fungi. This is the first report on efficient production of SmD1 uisng recombinant *E. coli* cells suitable for biologically active SmD1. The yield of defensin SmD1 was determined to be 1.21 μg/mL through quantitative Ni affinity chromatography elution of fusion protein. Moreover, recombinant protein was partially expressed as inclusion body. To reduce the formation of inclusion bodies and obtain the most activity production, the induction has been optimized. The diameter of the inhibition zone was chosen as the optimization standard because it is the most visual target for evaluating the antimicrobial activity of the recombination plant defensing (*Meng et al., 2016*).

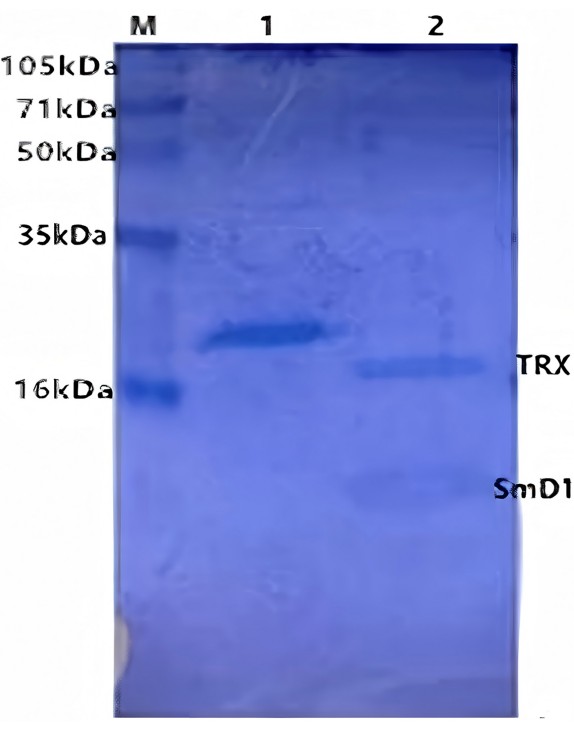

**Figure 4 SDS-PAGE of the products resulting from purification by Ni column.** M, polypeptide molecular mass standards; Lane 1, eluate from the Ni column with lysis buffer containing; Lane 2, proteins after formic acid treatment column.

## Optimization of induction condition

The induction conditions were optimized using *F. graminearum* CGMCC 3.6862 as the test strain. When OD600 is between 0.4 and 1.2, the inhibition zone increases first and then decreases (Fig. 5). It is inferred that a high initial bacterial concentration will lead to metabolic burden and affect the expression and stability of the target protein. When the $OD_{600}$ is 0.8, the diameter of the zone of inhibition reaches the maximum. The effect of induction time on the antifungal activity of the target protein is shown in Fig. 6. The strongest antifungal activity was obtained under 12 h induction.

To define the best induction temperature, four different induction temperatures (16, 25, 30 and 37 °C) were applied to steady-stated LB liquid. It showed that the highest antifungal activity emerged at the fraction from the target protein induced at 25 °C (Fig. 7). The target protein induced at temperature closer to 37 °C showed lowest activity, which may have accumulated too much inclusion, resulting in a high metabolic burden. When IPTG was between 0.2 and 1.2 IPTG, the inhibition zone increased first and then decreased. It is inferred that a high IPTG concentration will promote the generation of inclusion and have an adverse effect on cell growth and metabolism. When the IPTG concentration was 0.6 mM, the strongest antifungal activity was obtained.

The results showed that the antimicrobial activity of recombinant protein reached its maximum when the IPTG concentration was 0.6 mM; the induction temperature was 25 °C; the induction time was 12 h, and the $OD_{600}$ was 0.8.
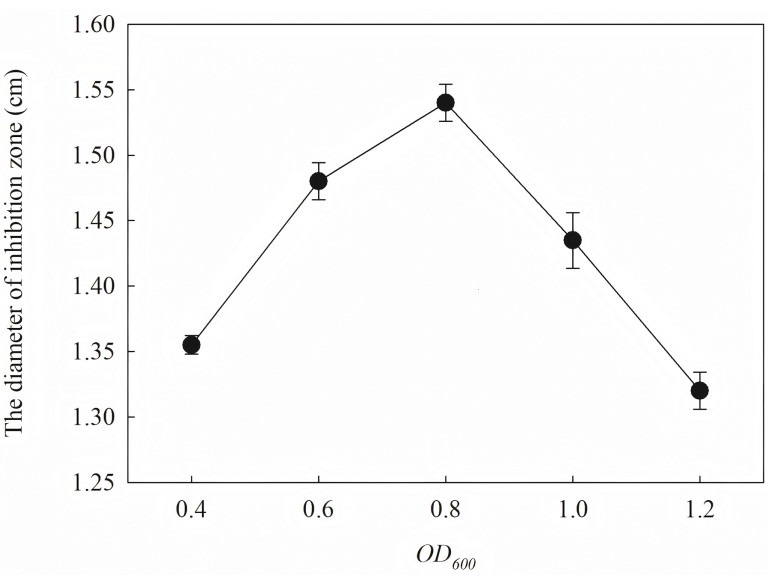

**Figure 5 Effect of different OD600 on antifungal activity.**

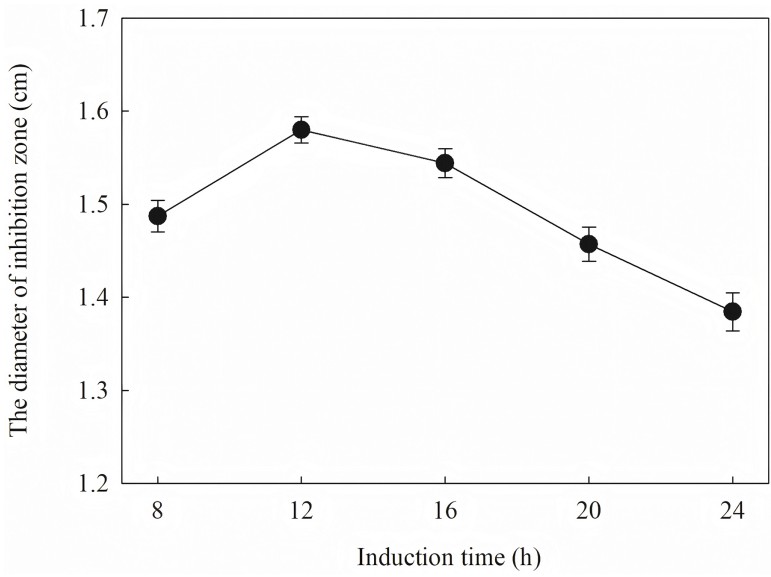

**Figure 6 Effect of different induction time on antifungal activity.**

Induction occurs at the logarithmic phase. After the addition of IPTG, the culture almost stops growing and begins to express recombination protein. Therefore, the number and activity of the culture are important for the induction. In optimization, the culture was induced when the $OD_{600}$ was 0.6, and the highest activity was obtained at an appropriate quantity. When the culture was induced earlier ($OD_{600}$ = 0.4) or later ($OD_{600}$ = 0.8), the

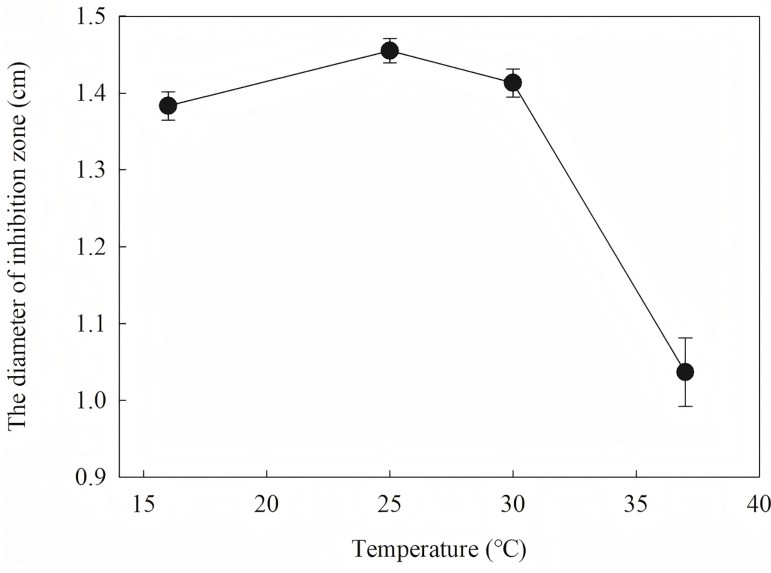

**Figure 7 Effect of different induction temperature on antifungal activity.**

final activity was lower due to an inappropriate of quantity or activity of bacteria. The induction time was also optimized. Due to IPTG being a strong inducer that can induce expression for a long time, enough induction time is required to obtain the maximum yield. However, the diameter of the inhibition zone continued to decrease after 12 h. The expression host *E. coil* has proteolytic enzyme that can degrade recombinant protein because it is a heterogenous protein. In another way, the recombination protein was almost soluble. Inclusion bodies are insoluble particles that cannot be degraded by the proteolytic. Inclusion bodies are unavoidable in the recombinant expression of *E. coil*. The efforts have been made to reduce the proportion of the inclusion bodies. Inclusion bodies were formed due to high expression and incorrect folding. The induction temperature has a great influence on the expression speed. Higher temperature (30 °C) reduced the antimicrobial activity for the formation of more inclusion bodies. When the temperature was at 37 °C, the antimicrobial activity decreased fast. Lower temperatures could reduce the formation of inclusion bodies, but the expression may slow down and cells may also be damaged at low temperatures. When the induction temperature was 25 °C, the production had the strongest antimicrobial activity. The concentration of the IPTG has an opposite influence on the recombinant expression and the formation of the inclusion bodies. Therefore, six concentrations of IPTG were detected to balance expression and inclusion bodies. As shown in Fig. 8, when the induction was carried out with 0.6 mM IPTG, the recombination SmD1 had the strongest antimicrobial activity.

## Antifungal activity assay

MIC and inhibition zone diameters were measured to assess the antifungal activity of the plant defensin SmD1. The MIC of SmD1 against four indictor phytopathogenic fungi,

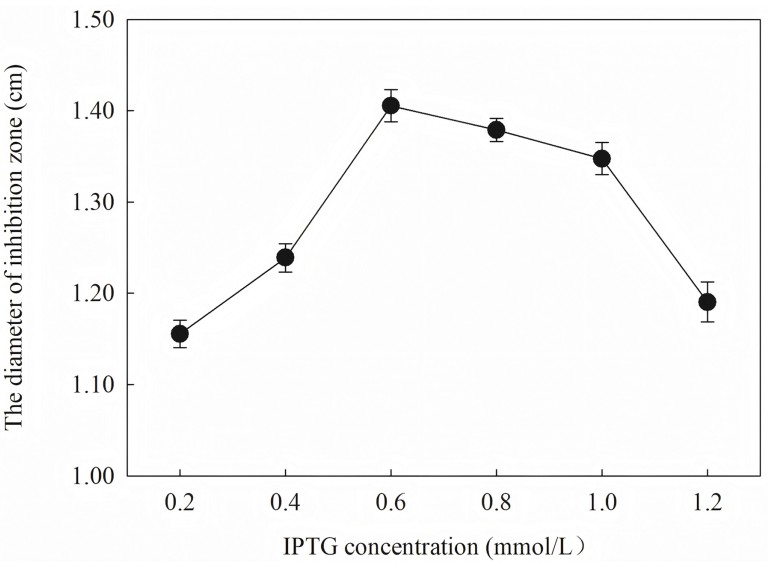

**Figure 8 Effect of different IPTG concentration on antifungal activity.**

| Table 1 Inhibition zone of defensin SmD1. | |
| --- | --- |
| The indicator strains | The diameter of inhibition zone (cm) |
| *B. cinereal* CGMCC 3.4584 | 1.59 |
| *F. graminearu* CGMCC 3.6862 | 1.52 |
| *F. oxysporum* CGMCC 3.2830 | 1.48 |
| *R. solani* CGMCC 3.2888 | 1.43 |

including *B. cinereal*, *F. graminearum*, *F. oxysporum* and *R. solani*, was determined by microdilution method. The MIC of plant defensin SmD1 against *B. cinereal* CGMCC3.4584 was 1 μg/mL, indicating the strongest antifungal activity against the four indicator phytopathogenic fungi detected. This is consistent with the strongest antifungal activity achieved against *B. cinereal* and *P. debaryanum* (*Slavokhotova et al., 2010*). The MIC of SmD1 against *F. graminearum* CGMCC3.6862 and *F. oxysporum* CGMCC3.2830, were both 8 μg/mL. The MIC of SmD1 against *R. solani* CGMCC3.288 was 4 μg/mL. In addition, SmD1 showed significant inhibition zones against four indictor phytopathogenic fungi detected, including *B. cinereal*, *F. graminearum*, *F. oxysporum* and *R. solani*, with the inhibition zone diameters ranging from 14.3 to 15.9 mm (Table 1).

Several benzimidazoles are used to control infections associated with *B. cinerea* in fruit and vegetables, but benzimidazoles resistance restricts further uses (*Raorane et al., 2022*). Some drugs such as quinoline alkaloids, azoles, isothiazoles, pyrimidines and pyridines were found to be effective against *R. solani*, *B. cinerea*, *F. graminearum*, *F. oxysporum* and *Sclerotinia sclerotiorum*, respectively (*An et al., 2023*). The antifungal activity of SmD1 against different plant pathogenic fungi makes it a potential substitute for chemical

fungicides in post-harvest preservation. Considering the further strategy of using these defensins to solve the problem of antibiotic resistance, we suggest integrating defensins with complementary antimicrobials into multi-mechanism formulations to reduce resistance evolution.

## CONCLUSION

This study establishes an optimized recombinant production platform for plant defensin SmD1, which solved toxicity constraints and yield limitations through systematic purification, cleavage and induction optimization. The resulting structurally intact and biologically active SmD1 demonstrates its efficacy as a next-generation plant fungicide, with a multi-target antibacterial mechanism that can combat the evolution of drug resistance. This work confirms that SmD1 is a promising biopharmaceutical grade plant protectant.

### Funding
This work was funded by the Research Foundation of the Education Department of Zhejiang Province, China, for the 2023 Annual Domestic Visiting Engineer Program on University-Enterprise Cooperation Projects (FG2023123), Agricultural Science and Technology Innovation Program of Institute of Food Science and Technology, Chinese Academy of Agricultural Sciences (CAAS-ASTIP-Q2024-IFST-09), and the Project of Drought-Alkali Wheat Processing Technology Innovation Center of Hebei Province (SJ2024032). The funders had no role in study design, data collection and analysis, decision to publish, or preparation of the manuscript.

### Grant Disclosures
The following grant information was disclosed by the authors:
Research Foundation of the Education Department of Zhejiang Province, China: FG2023123.
Institute of Food Science and Technology, Chinese Academy of Agricultural Sciences: CAAS-ASTIP-Q2024-IFST-09.
Project of Drought-Alkali Wheat Processing Technology Innovation Center of Hebei Province: SJ2024032.

### Author Contributions
- Yiyi Qiu conceived and designed the experiments, analyzed the data, prepared figures and/or tables, authored or reviewed drafts of the article, and approved the final draft.
- Qiaozhi Song performed the experiments, analyzed the data, authored or reviewed drafts of the article, and approved the final draft.

## Data Availability

The raw measurements are available in the Supplemental Files.

## Supplemental Information

Supplemental information for this article can be found online at http://dx.doi.org/10.7717/peerj.19526#supplemental-information.

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
