# Peer review of "Heterologous expression of the Stellaria media plant defensin SmD1 in Escherichia coli"

_PeerJ, doi:10.7717/peerj.19526_

## Round 0.1 · original submission · Major Revisions

Three referees assessed your manuscript. Two of them are positive about the content and one thinks the study is too basic. There are several recommendations that need to be attended. If so, I think the manuscript might be improved and have the chance to be moved to the next editorial stage. Please consider addressing the Reviewers' recommendations.

**Language Note:** The review process has identified that the English language must be improved. PeerJ can provide language editing services - please contact us at [email protected] for pricing (be sure to provide your manuscript number and title). Alternatively, you should make your own arrangements to improve the language quality and provide details in your response letter. – PeerJ Staff

Reviewer 1 ·

Basic reporting

The ms is simple.

Experimental design

The experiment has no novelty.

Validity of the findings

Due to the simple ms, I can not find the novelty.

Additional comments

I reviewed the ms, it expressed the SmD1 in E.coli and perform its ability to resistant against fungi, totally, it is simple and can not publish in the PeerJ. I am sorry I can not give you the positive response.
1. The introduction is not related to the research content. Please write the introduction around your own research content.
2. 3.1 has no meaning.
3. Fig2-4 were not clear.
4. Combine 5-8.

Reviewer 2 ·

Basic reporting

This study describes the heterologous expression of the Stellaria media plant defensin SmD1 in Escherichia coli BL21 (DE3). The SmD1 gene was fused to thioredoxin (TRX) to enhance solubility and was successfully expressed and purified using Ni-IDA chromatography. A formic acid cleavage strategy was employed to release active SmD1, and optimization of expression conditions was performed. The recombinant protein demonstrated antimicrobial activity against phytopathogenic fungi. The manuscript presents an important contribution to the production of plant defensins for potential agricultural applications.

Experimental design

The methods used for determining antimicrobial activity are not well described. Was the minimum inhibitory concentration (MIC) determined? Were any positive controls (e.g., commercially available antifungals or previously studied defensins) used for comparison?

The manuscript highlights E. coli as an efficient heterologous system but does not address why it was chosen over: 1) yeast expression systems (e.g., Pichia pastoris), which are better suited for disulfide bond-containing proteins; and 2) plant-based expression systems, which could provide better yield and correct folding. A brief discussion comparing different expression systems would strengthen the paper.

Validity of the findings

The yield of purified SmD1 should be explicitly reported.
How was the purity of the final protein assessed? Was mass spectrometry or HPLC used?
Given that SmD1 is cysteine-rich, were any redox conditions or refolding steps needed to ensure correct disulfide bond formation/function?

Given that plant defensins require correct disulfide bonding for activity, it is essential to confirm that SmD1 is properly folded. Circular dichroism (CD) spectroscopy to assess secondary structure or mass spectrometry to confirm correct molecular weight and integrity would be highly recommended.

Additional comments

The study addresses the challenge of low plant defensin production by using an E. coli-based expression system, with recombinant SmD1 showing strong antifungal activity for potential plant disease control. The TRX fusion system improves solubility, and expression conditions were systematically optimized for reproducibility. Functional validation confirms that recombinant SmD1 is bioactive against multiple phytopathogenic fungi.

·

Basic reporting

In the article, they authors heterologously produce a plant defensin in E. coli and test its antifungal activity. However, there are some suggestions to increase the quality of the paper:
- Authors should be careful when writing temperatures; they should use 30 °C instead of incorrect formats.
- Scientific names should be written in italics.
- Authors should be careful not to confuse antibacterial with antifungal activity (Section 3.5).
- Authors should be careful with the names of the fungi used, e.g. B. cinerea and F. graminearum
- All bibliographic references must follow the same format.
-References should be more up-to-date, e.g., Brian and Bray (2003).
-It is recommended that the discussion of results be supported by bibliographical references.

Experimental design

Overall, the experimental design is adequate; however, some additions are suggested below for improvement.
- Clarify that antifungal tests are performed, not antibacterial tests.
- Line 77: How much protein is there in 100 uL?
-Figure 3: Indicate the band corresponding to the described results.
- It is suggested that antifungal assays include a growth inhibition control, as well as a description of the size of the 'n', as well as the statistic test employed (Table 1, and Figures 5-8).
-Please indicate the strain used in the assays in Figures 5-8.

Validity of the findings

I believe that growth and inhibition controls are necessary in antifungal assays, as well as the indication of the strain being used. A robust statistical analysis is needed to support the authors' suggestion.

Additional comments

-It is recommended that you indicate whether the fungal strains are resistant to any antifungal agents, as this would broaden the scope of the study.
-Line 31 and 32: improve writing.
-Line 52: Correct the bibliographic reference.
-Line 77: indicate the name of the gel extraction kit.

---

## Round 0.2 · accepted · Accept

The Reviewers consider that the manuscript has been significantly improved and is now suitable for publication.

Reviewer 1 ·

Basic reporting

no comment

Experimental design

no comment

Validity of the findings

no comment

Additional comments

no comment

Reviewer 2 ·

Basic reporting

After reviewing the revised version, I believe it meets the necessary standards and is well-prepared for publication. I have no further suggestions or changes at this point.

Experimental design

After reviewing the revised version, I believe it meets the necessary standards and is well-prepared for publication. I have no further suggestions or changes at this point.

Validity of the findings

After reviewing the revised version, I believe it meets the necessary standards and is well-prepared for publication. I have no further suggestions or changes at this point.